# A retrospective analysis to estimate the healthcare resource utilization and cost associated with treatment-resistant depression in commercially insured US patients

**Gang Li** [1]*, **Ling Zhang**[1¤], **Allitia DiBernardo**[2], **Grace Wang**[1], **John J. Sheehan**[3], **Kwan Lee**[1], **Johan Reutfors**[4], **Qiaoyi Zhang**[1]

1 Janssen Research & Development, LLC, Raritan, New Jersey, United States of America, 2 Janssen Research & Development, LLC, Titusville, New Jersey, United States of America, 3 Janssen Scientific Affairs, LLC, Titusville, New Jersey, United States of America, 4 Karolinska Institutet, Stockholm, Sweden

¤ Current address: Boehringer Ingelheim Pharmaceuticals, Inc, Ridgefield, Connecticut, United States of America
* GLi@its.jnj.com

**Data Availability Statement:** The data for these analyses were made available to the authors by third-party license from Optum, a commercial data

## Abstract

### Objective

The economic burden of commercially insured patients in the United States with treatment-resistant depression and patients with non–treatment-resistant major depressive disorder was compared using data from the Optum Clinformatics™ claims database.

### Methods

Patients 18–63 years on antidepressant treatment between 1/1/13 and 9/30/13, who had no treatment claims for depression 6 months before the index date (first antidepressant dispensing), and who had a major depressive disorder or depression diagnosis within 30 days of the index date, were included. Treatment-resistant depression was defined as receiving 3 antidepressant regimens during 1 major depressive disorder episode. Patients with treatment-resistant depression were matched with patients with non–treatment-resistant major depressive disorder at a 1:4 ratio using propensity score matching. The study consisted of 1-year baseline (pre-index) and 2-year follow-up (post index) periods. Cost outcomes were compared using a generalized linear model.

### Results

2,370 treatment-resistant depression and 9,289 non–treatment-resistant major depressive disorder patients were included. In year 1 of the follow-up period, compared with non–treatment-resistant major depressive disorder, patients with treatment-resistant depression had: more emergency department visits (odds ratio = 1.39, 95% confidence interval = 1.24–1.56); more inpatient hospitalizations (odds ratio = 1.73, 95% confidence interval = 1.46–2.05); longer hospital stays (mean difference vs non–treatment-resistant major depressive

provider in the US, and Janssen Pharmaceuticals (who have a license for analysis of the Optum Clinformatics™ Extended Data Mart [CEDM]). As such, the authors cannot provide the raw data themselves. Other researchers could access the data by purchase through Optum; and the inclusion criteria specified in the Methods section would allow them to identify the same cohort of patients we used for these analyses. Interested individuals may visit https://www.optum.com/solutions/life-sciences.html for more information on accessing Optum CEDM data. We confirm that no authors had special privileges to access data from Optum via third-party license, and that other researchers would be able to access the data in the same manner as the authors.

**Funding:** Janssen Research & Development, LLC provided funding for this study in the form of salaries for GL, LZ, AD, GW, JJS, KL, and QZ, as well as funding for editorial support. Schering-Plough also provided funding in the form of unrestricted grant support to JR. The specific roles of these authors are articulated in the 'author contributions' section. The funders had no role in study design, data collection and analysis, decision to publish, or preparation of the manuscript.

**Competing interests:** The authors have read the journal's policy and have the following competing interests: GL, AD, GW, JS, KL, and QZ are employees of Janssen Research & Development, LLC and hold stock in the company. LZ was an employee of Janssen Research & Development, LLC, at the time this study took place and is currently an employee of Boehringer Ingelheim Pharmaceuticals, Inc. JR is in research collaboration with Janssen, AstraZeneca, AbbVie, and Pfizer, for which Karolinska Institutet has received grant support. JR has been a speaker for Eli Lilly and received unrestricted grant support from Schering-Plough. Wilson Joe, PhD, of MedErgy, provided editorial support for this manuscript. Editorial support was funded by Janssen Research & Development, LLC. This does not alter our adherence to PLOS ONE policies on sharing data and materials. There are no patents, products in development or marketed products associated with this research to declare.

disorder = 2.86, 95% confidence interval = 0.86–4.86 days); and more total healthcare costs (mean difference vs non–treatment-resistant major depressive disorder = US$3,846, 95% confidence interval = $2,855-$4,928). These patterns remained consistent in year 2 of the follow-up period.

## Conclusion

Treatment-resistant depression was associated with higher healthcare resource utilization and costs versus non–treatment-resistant major depressive disorder in this commercially insured cohort of patients in the United States.

## Introduction

Depression is a widespread, severely disabling disorder associated with impaired daily functioning, diminished quality of life, and increased mortality and healthcare utilization [1–4]. Healthcare costs such as outpatient medical services, pharmaceutical services, and inpatient services as well as indirect costs such as workplace presenteeism and absenteeism all contribute substantially towards the total burden of major depressive disorder [3]. In 2012, the US societal economic burden of major depressive disorder was estimated at $188 billion, which exceeded the US societal burden of cancer ($131 billion) and diabetes ($173 billion) [4].

The goal of major depressive disorder treatment is to achieve remission, a subclinical state where the patient is no more than mildly symptomatic, fully functional, and essentially indistinguishable from those without major depressive disorder [5–7]. However, as measured by the Sequenced Treatment Alternatives to Relieve Depression (STAR*D) study, which used a large, representative patient sample and presented a comprehensive view of nonresponse to depression treatment, approximately 30% of patients with major depressive disorder do not achieve remission even after adequate trials of 2 antidepressant treatments [5]. The term "treatment-resistant depression" refers to depression that does not respond to antidepressant therapy [6]. Although no consensus definition currently exists, the US Agency for Healthcare Research and Quality (AHRQ) and Food and Drug Administration (FDA) proposed a standard definition of treatment-resistant depression: failing to respond to a minimum of 2 antidepressants administered at an adequate dose, for an adequate duration [8,9]. There is currently no consensus regarding the definitions of 'adequate'; a recent review on treatment-resistant depression found that most studies considered an adequate treatment duration to last for a minimum of 4 or 6 consecutive weeks, with the majority requiring ≥4 weeks [10,11]. In the current study, we defined adequate dose based upon the American Psychiatric Association Practice Guidelines for Treatment of Patients with Major Depressive Disorder [12] and approved recommended minimal dosage, and adequate duration was defined by an algorithm that required ≥29 days of prescription coverage (details outlined in Materials and Methods below). Results from STAR*D suggest that nonresponse to 2 adequate trials of established pharmacotherapy classes is an inflection point that predicts a poor prognosis with respect to low remission and high relapse rates, and is associated with higher rates of future medication intolerance [5,13].

Greater healthcare utilization and higher healthcare costs have been demonstrated in patients with treatment-resistant depression compared with non–treatment-resistant major depressive disorder [14–22]. According to three recent estimates, per-patient per-year direct healthcare costs in patients with treatment-resistant depression versus non–treatment-resistant major depressive disorder were US$6,709 higher among commercially-insured patients,

$4,382 higher among Medicaid-insured patients, and $9,479 higher among US integrated delivery network–insured patients [14,20,22]. Among commercially-insured patients, indirect work loss–related costs were also US$1,811 greater in patients with treatment-resistant depression [14]. Although both treatment resistance and symptom severity are associated with increased direct and indirect costs in patients with major depressive disorder, treatment-resistant depression appears to be the primary contributor to the economic burden of depression [23]. This retrospective study was conducted to provide an updated estimate of the economic burden of patients with treatment-resistant depression compared with patients with non–treatment-resistant major depressive disorder using recent US data and a more comprehensive definition of treatment-resistant depression.

## Materials and methods

### Data source

This study was based on insurance claims data from the Optum Clinformatics™ Extended Data Mart (CEDM). CEDM stores medical and pharmacy benefit coverage records of commercial and Medicare Advantage health plan members. Data are routinely captured, verified, automated, and de-identified, providing a key information source for various research efforts.

### Sample selection and study design

The study sample included patients, 18 to 63 years old, who had an antidepressant medication dispensed between January 1, 2013 and September 30, 2014 and had no claims for pharmacologic or nonpharmacologic depression treatments 6 months before the index date (the date of first antidepressant medication dispensing). The 12 months before the index date constituted the baseline period, and the 24 months after the index date constituted the follow-up period. Patients included in the analysis had a clinician's diagnosis of major depressive disorder according the UnitedHealthcare Guidelines (*International Classification of Diseases, 9th Edition* [ICD-9] 296.2x and 296.3x [except for 296.25 and 296.30], 300.4, 309.0, 309.1, and 311) within ±30 days of the index date [24]. Eligible patients had to have ≥2 consecutive antidepressant medications dispensed (with a gap of ≤30 days after the index date for the 3 non–major depressive disorder diagnosis categories) to ensure some level of compliance. Except for patients who died during the study period, all patients were continuously enrolled in the health plan, for both pharmacy and medical benefits, during the baseline and follow-up periods.

Patients were excluded if they had an ICD-9 or *International Classification of Diseases, 10th Edition* (ICD-10) code for psychosis, schizophrenia, mania, or bipolar disorder, or an ICD-9 code for dementia at any time. Additionally, patients were excluded if they had a dispense of lithium, thyroid hormone (T3 or T4), an antipsychotic, or an anti-epileptic–type mood stabilizer, or if they received electroconvulsive treatment or transcranial magnetic stimulation during the 6 months before the index date.

Use of the database was reviewed by the New England Institutional Review Board (IRB) and was determined to be exempt from IRB approval, as this study did not involve human subjects research.

### Treatment-resistant depression

Patients were identified as having treatment-resistant depression or non–treatment-resistant major depressive disorder using a claims-based algorithm. This study employed a definition of treatment-resistant depression based upon the AHRQ definition: depression that fails to respond to a minimum of 2 antidepressant treatments administered at an adequate dose and

duration (referred to as drugs A and B below) [9]. A listing of treatments that were defined as antidepressants is shown in S1 Table. Accordingly, a patient with major depressive disorder was considered to have treatment-resistant depression if the patient received 3 antidepressant regimens, of adequate dose and duration for the first 2, in the current major depressive disorder episode. The first regimen was required to be an antidepressant, but the second and third regimens could be an antidepressant taken alone, with another antidepressant, or with an augmentation medication (anticonvulsant, antipsychotic, lithium, psychostimulant, or thyroid hormone) [12,25].

An adequate antidepressant dose was defined by the recommended minimal dosage in the American Psychiatric Association major depressive disorder practice guidelines [12] or in the US Food and Drug Administration-approved package inserts. Adequate duration was assessed using an algorithm to determine medication failure of drug A (first-line treatment) based on its treatment duration before the introduction of drug B. Drug A was considered a failure if drug B was introduced between 29 and 180 days or if drug A was augmented with drug B starting on Day 15 or later; the same algorithm was used to determine the failure of drug B based on the introduction of drug C.

## Assessments

Patient demographic and baseline clinical characteristics were assessed and compared between treatment-resistant depression and non–treatment-resistant depression groups. Characteristics included: age group (18–24, 25–34, 35–44, 45–54, and 55–63 years); sex; index year (2013/2014); depression diagnosis (296.2, 296.3, 300, 309, or 311) within 30 days of the index date; diagnosis of anxiety, substance abuse, personality disorder, and post-traumatic stress disorder (PTSD) during the baseline period; and Elixhauser comorbidity score, calculated using diagnosis codes during the baseline period [26]. A propensity score was derived from the age on index date; gender; depression diagnosis code around index date; baseline diagnosis of anxiety, personality disorder, substance abuse, and/or PTSD; and Elixhauser comorbidity score. Patients with treatment-resistant depression were matched to those with non–treatment-resistant major depressive disorder using the propensity score at a 1:4 ratio with the greedy approach and calipers of width equal to 0.02.

Healthcare utilization and costs were estimated annually for 2 consecutive years during the follow-up period. Number of outpatient visits (which included office based and ambulatory hospital outpatient visits), proportion of patients with emergency department (ED) visits, proportion of patients with hospitalizations, and hospital length of stay (LOS; ie, the sum of hospital stay days from all hospitalizations during the one-year follow-up period) were assessed to measure resource utilization. Costs were estimated from the payer and patient perspectives. Medical costs to payers included claims for outpatient visits, ED visits, and hospitalizations; pharmacy costs to payers were the sum of pharmacy claims; and total costs to payers were the sum of medical costs and pharmacy costs to payers. Medical costs to patients were defined as the sum of deductibles, copayments, and coinsurance for all medical services; prescription costs to patients were defined as the sum of deductibles, copayments, and coinsurance for all prescription drugs; and total costs to patients were the sum of medical costs and prescription costs to patients. Total healthcare costs were defined as the sum of costs to payers and patients. All cost estimates were made using 2017 US$ rates.

## Statistical analysis

Descriptive statistics were generated to summarize patient characteristics and outcome measures for treatment-resistant depression and non–treatment-resistant depression cohorts.

Between-group comparisons of demographic and baseline clinical characteristics were made using chi-square tests for categorical variables and t-tests for continuous variables.

Annual resource utilization and costs associated with care for patients with treatment-resistant depression versus non–treatment-resistant major depressive disorder were compared using a generalized linear model and log-link function with negative binomial distribution for resource utilization and gamma distribution for cost (SAS GENMOD procedure), adjusted for the baseline value of the variable [12]. Estimates and 95% confidence intervals (CIs) were obtained using bootstrapping with 1000 iterations. In a sensitivity analysis, cost data were analyzed using a similar linear model with normal distribution. Additionally, a linear model with normal distribution was used to calculate differences in costs for patients with treatment-resistant depression versus non–treatment-resistant major depressive disorder, adjusting for the baseline variable, by depression diagnosis codes (major depressive disorder, ICD-9 296.X; dysthymic disorder, ICD-9 300.X; adjustment disorder, ICD-9 309.X; depressive disorder not otherwise specified, ICD-9 311.X). The odds ratio (OR) of hospitalization and ED visits for patients with treatment-resistant depression versus non–treatment-resistant major depressive disorder was estimated using a logistic regression model with repeated measurements (SAS GENMOD procedure) adjusted for the respective baseline value of the variable, year of follow-up, treatment-resistant depression status * year, and baseline Elixhauser score [27].

Two high dimensional covariate selection approaches [28] were implemented as part of sensitivity analyses to identify covariates that might potentially impact costs in addition to those pre-specified for the propensity score matching (see S1 Appendix for details).

## Results

### Patient disposition and baseline characteristics

Of 17,859 eligible patients diagnosed with major depressive disorder, 2,384 (13%) had treatment-resistant depression and 15,475 (87%) had non–treatment-resistant major depressive disorder (see patient disposition flow diagram in S1 Fig). Compared with patients with non–treatment-resistant major depressive disorder, patients with treatment-resistant depression were slightly younger, more likely to be female, and to have a history of anxiety or PTSD. After propensity score matching, 2,370 patients with treatment-resistant depression and 9,289 patients with non–treatment-resistant major depressive disorder were included in the analysis. Mean age after matching was 39.2 years, and 62% of patients were female. Baseline characteristics of patients in the 2 groups were comparable, except for the Elixhauser score (Table 1); therefore, Elixhauser score was adjusted in the healthcare utilization and cost analyses.

### Healthcare utilization

Healthcare utilization was significantly and consistently higher in the treatment-resistant depression group than in the non–treatment-resistant major depressive disorder group (Table 2). In the first year of follow up, compared with non–treatment-resistant major depressive disorder, patients with treatment-resistant depression had: more emergency department visits (OR = 1.39, 95% CI = 1.24–1.56]) and more inpatient hospitalizations (OR = 1.73, 95% CI = 1.46–2.05]). In addition, the difference in adjusted predicted outcomes between patients with and without treatment-resistant depression was a 2.86-day longer length of hospital stay (difference vs non–treatment-resistant major depressive disorder, 95% CI = 2.86, 0.86–4.86 days) and 2.95 more outpatient visits (difference vs non–treatment-resistant major depressive disorder, 95% CI = 2.95, 2.48–3.43 visits). These patterns remained in the second year of follow up.

**Table 1. Demographic and clinical characteristics of patients with treatment-resistant depression and non–treatment-resistant depression before and after the propensity score matching.**

| Characteristic | Unmatched | | | | | Matched | | | | |
|---|---|---|---|---|---|---|---|---|---|---|
| | Treatment-resistant depression (N = 2,384) | | Non–treatment-resistant major depressive disorder (N = 15,475) | | | Treatment-resistant depression (n = 2,370) | | Non–treatment-resistant major depressive disorder (n = 9,289) | | |
| | n | % | n | % | P | n | % | n | % | P |
| Age (years) | Mean 39.2 | SD 13.0 | Mean 40.1 | SD 12.9 | 0.0021 | Mean 39.2 | SD 12.9 | Mean 39.2 | SD 12.8 | 0.9635 |
| Age group (years) | | | | | | | | | | |
| 18–24 | 472 | 20 | 2724 | 18 | | 468 | 20 | 1746 | 19 | |
| 25–34 | 440 | 18 | 2779 | 18 | | 437 | 18 | 1770 | 19 | |
| 35–44 | 585 | 25 | 3735 | 24 | | 582 | 25 | 2335 | 25 | |
| 45–54 | 547 | 23 | 3657 | 24 | | 545 | 23 | 2132 | 23 | |
| 55–63 | 340 | 14 | 2580 | 17 | 0.0082 | 338 | 14 | 1306 | 14 | 0.8111 |
| Female sex | 1481 | 62 | 9209 | 60 | 0.0154 | 1474 | 62 | 5758 | 62 | 0.8531 |
| Comorbidities | | | | | | | | | | |
| Anxiety | 659 | 28 | 3664 | 24 | < .0001 | 646 | 27 | 2392 | 26 | 0.6138 |
| Personality Disorder | 17 | 1 | 53 | <1 | 0.0813 | 8 | <1 | 9 | <1 | 0.7099 |
| Substance Abuse | 73 | 3 | 392 | 3 | 0.1056 | 71 | 3 | 227 | 2 | 0.1217 |
| PTSD | 37 | 2 | 179 | 1 | < .0001 | 35 | 1 | 113 | 1 | 0.2195 |
| Elixhauser score[a] | Mean 1.8 | SD 1.35 | Mean 1.70 | SD 1.26 | 0.0645 | Mean 1.74 | SD 1.33 | Mean 1.68 | SD 1.21 | 0.0293 |
| Major depressive disorder diagnostic code | | | | | | | | | | |
| Major depressive disorder (ICD-9 296.X) | 1003 | 42 | 5485 | 35 | | 991 | 42 | 3831 | 41 | |
| Dysthymic disorder (ICD-9 300.X) | 323 | 14 | 2307 | 15 | | 322 | 14 | 1235 | 13 | |
| Adjustment disorder (ICD-9 309.X) | 65 | 3 | 520 | 3 | | 65 | 3 | 205 | 2 | |
| Depressive disorder NOS (ICD-9 311.X) | 993 | 42 | 7163 | 46 | < .0001 | 992 | 42 | 4018 | 43 | 0.3248 |

CI, confidence interval; SD, standard deviation.

[a]The Elixhauser score ranges from 0 to 30.

**Table 2. Healthcare resource utilization per year during the study period.**

| Variable | Treatment-resistant depression | | Non–treatment-resistant major depressive disorder | | Treatment-resistant depression vs non–treatment-resistant major depressive disorder | | |
|---|---|---|---|---|---|---|---|
| | n | % | n | % | Odds ratio | 95% CI | |
| Patients with ED visits in Year 1, % | 628 | 26 | 2933 | 19 | 1.39 | 1.24 | 1.56 |
| Patients with ED visits in Year 2, % | 530 | 22 | 2714 | 18 | 1.27 | 1.13 | 1.43 |
| Patients with inpatient hospitalization in Year 1, % | 201 | 8 | 754 | 5 | 1.73 | 1.46 | 2.05 |
| Patients with inpatient hospitalization in Year 2, % | 163 | 7 | 737 | 5 | 1.43 | 1.19 | 1.73 |
| Variable | Treatment-resistant depression | | Non–treatment-resistant major depressive disorder | | Treatment-resistant depression vs non–treatment-resistant major depressive disorder | | |
| | n | | n | | Estimate of mean difference | 95% CI | |
| Hospital LOS in Year 1, number of days | 8.75 | | 5.90 | | 2.86 | 0.86 | 4.86 |
| Hospital LOS in Year 2, number of days | 9.60 | | 6.19 | | 3.41 | −0.43 | 7.25 |
| Number of outpatient visits in Year 1 | 11.45 | | 8.50 | | 2.95 | 2.48 | 3.43 |
| Number of outpatient visits in Year 2 | 7.39 | | 5.59 | | 1.79 | 1.35 | 2.24 |

CI, confidence interval; ED, emergency department; LOS, length of stay.

## Healthcare costs

Consistent with the increased healthcare utilization observed in patients with treatment-resistant depression, costs were significantly higher in the treatment-resistant depression group (Table 3). The adjusted mean (95% CI) differences in total payer costs between the treatment-resistant depression and the non–treatment-resistant major depressive disorder groups were US$3,430 ($2,438-$4,478) for year 1 and US$2,191 ($1,031-$3,453) for year 2. Estimated between-group mean (95% CI) differences in patients' total out-of-pocket costs were US$354 ($260-$457) for year 1 and US$184 ($91-$285) for year 2. For total healthcare costs, including both reimbursed costs and costs to patients, estimated between-group mean (95% CI) differences were US$3,846 ($2,855-$4,928) in year 1 and US$2,412 ($1,217-$3,713) in year 2. The results from the linear model with normal distribution were consistent: treatment-resistant depression patients had statistically significantly higher reimbursed costs as well as costs to patients (see S2 Table). The results from the two high dimensional covariate selection approaches that adjusted for additional covariates were also consistent (see S1 Appendix and tables and figure therein). Mean cost differences between patients with treatment-resistant depression versus non–treatment-resistant depression varied based on depression diagnosis code, but differences in sample sizes limit interpretation (see S3 Table).

## Discussion

This study assessed healthcare utilization and costs of treatment-resistant depression, analyzing data from a US patient sample obtained from the CEDM database. During year 1 and year

**Table 3. Comparison of costs per year between treatment-resistant depression and non–treatment-resistant major depressive disorder patients (US$)[a].**

| Variable | Treatment-resistant depression | Non–treatment-resistant major depressive disorder | Treatment-resistant depression vs non–treatment-resistant major depressive disorder | | |
|---|---|---|---|---|---|
| | | | Adjusted mean difference | 95% CI | |
| **Cost to payers** | | | | | |
| Medical cost in Year 1 | 9075 | 6125 | 2950 | 2051 | 3978 |
| Medical cost in Year 2 | 8393 | 6621 | 1772 | 632 | 2958 |
| Pharmacy cost in Year 1 | 2043 | 1507 | 535 | 300 | 789 |
| Pharmacy cost in Year 2 | 2027 | 1664 | 362 | 58 | 720 |
| Total cost to payers in Year 1 | 11014 | 7585 | 3430 | 2438 | 4478 |
| Total cost to payers in Year 2 | 10175 | 7984 | 2191 | 1031 | 3453 |
| **Cost to patients** | | | | | |
| Medical cost in Year 1 | 1373 | 1019 | 444 | 347 | 556 |
| Medical cost in Year 2 | 1207 | 1022 | 245 | 150 | 344 |
| Prescription cost in Year 1 | 406 | 318 | 88 | 68 | 109 |
| Prescription cost in Year 2 | 350 | 301 | 49 | 30 | 70 |
| Total cost to patients in Year 1 | 1767 | 1323 | 354 | 260 | 457 |
| Total cost to patients in Year 2 | 1499 | 1254 | 184 | 91 | 285 |
| **Total healthcare cost** | | | | | |
| Total healthcare cost in Year 1 | 12726 | 8881 | 3846 | 2855 | 4928 |
| Total healthcare cost in Year 2 | 11591 | 9179 | 2412 | 1217 | 3713 |

CI, confidence interval.

[a]Medical costs to payers included claims for outpatient visits, ED visits, and hospitalizations; pharmacy costs to payers were the sum of pharmacy claims; and total costs to payers were the sum of medical costs and pharmacy costs to payers. Medical costs to patients were defined as the sum of deductibles, copayments, and coinsurance for all medical services; prescription costs to patients were defined as the sum of deductibles, copayments, and coinsurance for all prescription drugs; and total costs to patients were the sum of medical costs and prescription costs to patients. Total healthcare costs were defined as the sum of costs to payers and patients.

2 following the index date, healthcare utilization was significantly higher in the treatment-resistant depression group than in the non–treatment-resistant major depressive disorder group. Consistent with this finding, patients with treatment-resistant depression had significantly higher reimbursed and out-of-pocket medical, pharmacy, and total healthcare costs.

Multiple prior studies of treatment-resistant depression have employed a range of criteria to define the treatment-resistant depression patient population. These criteria typically include some combination of the following: clinical diagnosis of depression, number of treatments used (>2 to ≥4), use of specific medications, time on medication(s), upward titration of medication(s), use of optimization strategies, and results from questionnaires [14–22,29]. Therefore, it is not surprising to observe the variance in reported percentages of patients with major depressive disorder who were treatment-resistant, which ranged between 11% and 30%; in most studies, patients with treatment-resistant depression were predominately female (64% to 74%) and between 35 and 55 years of age, although in 1 study [17] only 41% of patients with treatment-resistant depression were female. Given the lack of a consensus treatment-resistant depression definition, we adopted an evidence-based, comprehensive definition based on the AHRQ definition employed in the STAR*D trial: failure to respond to 2 oral antidepressant treatments of adequate duration and dose.

In this study, mean total healthcare costs to payers in year 1 and year 2, respectively, were 45% and 27% higher for patients with treatment-resistant depression compared with those with non–treatment-resistant major depressive disorder. This result is consistent with prior work across a range of patient populations, which found a 25%-134% higher burden among those with treatment-resistant depression versus those with non–treatment-resistant major depressive disorder [14–22]. Although not absolute, in general those analyses that assessed some component(s) of the indirect burden of major depressive disorder, such as productivity, identified larger percentage increases in the burden, suggesting that the incremental indirect burden of major depressive disorder among those with treatment-resistant depression is larger than the incremental direct burden.

One unexpected finding of the current is study is that estimated differences in healthcare utilization and costs between the treatment-resistant depression and non–treatment-resistant major depressive disorder groups were generally smaller in year 2 than in year 1. Specifically, we found that the number of outpatient visits and proportions of patients who had ED visits and inpatient hospitalization were all reduced during year 2 compared with year 1 for both treatment-resistant depression and non–treatment-resistant major depressive disorder groups, and the estimated mean differences were also slightly reduced. Although hospital LOS did not follow that trend, the estimated mean difference between treatment-resistant depression and non–treatment-resistant major depressive disorder groups was not significant in year 2. As episodes of treatment-resistant depression are generally longer than those of non–treatment-resistant major depressive disorder [30], it was anticipated that the incremental burden of treatment-resistant depression would remain constant in year 2 or increase compared with year 1. However, the episodic nature of major depressive disorder, or the possibility that over time and multiple medication changes switches, patients may eventually find an effective treatment may help to explain this relative decrease; however, it is unclear whether these year-to-year differences are clinically meaningful and further investigation is needed.

In contrast to most previous studies, which focused on healthcare costs reimbursed by payers, the current study also examined patients' out-of-pocket costs. Patients with treatment-resistant depression had out-of-pocket medical and pharmacy costs of US$1,323 in year 1 and US$1,254 in year 2. Compared with non–treatment-resistant major depressive disorder, costs for treatment-resistant depression represented increases of US$354 in year 1 and US$184 in year 2. These costs are likely to represent a substantial burden for many patients with

treatment-resistant depression. In the STAR*D study, participants reported high unemployment rates, ranging from 36% for patients who responded to step 1 treatment to 47% for patients who progressed to step 4 [5]. In another STAR*D analysis, patients with treatment-resistant depression demonstrated lower vocational productivity than patients with non–treatment-resistant major depressive disorder [31]. A claims-based study found that employees with treatment-resistant depression had an average of 35.8 work loss days per year, which was 1.7 times the rate of work loss days in employees with non–treatment-resistant major depressive disorder and 6.2 times that of those without major depressive disorder [14]. Thus, the higher out-of-pocket healthcare costs associated with treatment-resistant depression shown in the current analysis may represent a considerable financial hardship for this vulnerable population.

This study has several limitations. Data were from a claims database, which captures diagnoses recorded for reimbursement purposes rather than clinical diagnoses. Depression may be underreported in claims data for various reasons such as social stigma and financial incentives to bill for general medical disorder management. Diagnoses were based on individual physicians' clinical judgment and did not receive additional validation. Medication changes suggest treatment failure, but it is not possible to disentangle switches due to lack of efficacy or tolerability, or patient choice. In the absence of full medical histories, patients' major depressive disorder previous history, such as years of diagnosed major depressive disorder and number of major depressive disorder episodes, was not captured. Furthermore, results obtained using the Optum Clinformatics™ database may not generalize beyond patients with employer-sponsored commercial insurance and Medicare Advantage insurance.

## Conclusions

The results of this retrospective study suggest that patients with treatment-resistant depression have significantly greater healthcare utilization than matched patients with non–treatment-resistant major depressive disorder. This difference in healthcare utilization translates into significantly higher reimbursed and out-of-pocket medical, pharmacy, and overall costs for patients with treatment-resistant depression than those with non–treatment-resistant major depressive disorder.

## Supporting information

**S1 Appendix. High dimensional covariate selection approaches.**
(DOCX)

**S1 Fig. Patient disposition.**
(DOCX)

**S1 Table. List of antidepressant medications and minimum adequate dose.**
(DOCX)

**S2 Table. Comparison of costs per year (US$) between treatment-resistant depression and non–treatment-resistant major depressive disorder patients by the model with gamma log link vs a linear model[a].**
(DOCX)

**S3 Table. Difference in least square means of costs per year (US$) between treatment-resistant depression and non–treatment-resistant major depressive disorder patients by depression diagnosis codes (from linear models)[a].**
(DOCX)

**S4 Table. Demographic and clinical characteristics of patients with treatment-resistant depression and non–treatment-resistant depression before and after the propensity score matching at 1:1 ratio.**
(DOCX)

**S5 Table. Healthcare resource utilization per year during the study period (matched at 1:1 ratio).**
(DOCX)

**S6 Table. Comparison of costs per year between treatment-resistant depression and non–treatment-resistant major depressive disorder patients (US$; matched at 1:1 ratio)[a].**
(DOCX)

## Author Contributions

**Conceptualization:** Gang Li, Allitia DiBernardo, Grace Wang, Johan Reutfors, Qiaoyi Zhang.

**Formal analysis:** Gang Li, Ling Zhang, Grace Wang, Kwan Lee, Qiaoyi Zhang.

**Investigation:** Gang Li, Ling Zhang, Allitia DiBernardo, Grace Wang, John J. Sheehan, Kwan Lee, Johan Reutfors, Qiaoyi Zhang.

**Methodology:** Gang Li, Ling Zhang, Allitia DiBernardo, Grace Wang, John J. Sheehan, Kwan Lee, Johan Reutfors, Qiaoyi Zhang.

**Validation:** Gang Li, Ling Zhang, Grace Wang.

**Visualization:** Gang Li, Ling Zhang, Allitia DiBernardo, Grace Wang, John J. Sheehan, Kwan Lee, Johan Reutfors, Qiaoyi Zhang.

**Writing – original draft:** Gang Li, Ling Zhang, Allitia DiBernardo, Grace Wang, John J. Sheehan, Kwan Lee.

**Writing – review & editing:** Gang Li, Allitia DiBernardo, John J. Sheehan, Johan Reutfors, Qiaoyi Zhang.

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
