## [Decision Letter · Decision Letter 0]

25 Mar 2020

PONE-D-19-34560

A retrospective analysis to estimate the healthcare resource utilization and cost associated with treatment-resistant depression

PLOS ONE

Dear Dr. Li,

Thank you for submitting your manuscript to PLOS ONE. After careful consideration, we feel that it has merit but does not fully meet PLOS ONE’s publication criteria as it currently stands. Therefore, we invite you to submit a revised version of the manuscript that addresses the points raised during the review process. Reviewer comments are included below.

We would appreciate receiving your revised manuscript by May 09 2020 11:59PM. To enhance the reproducibility of your results, we recommend that if applicable you deposit your laboratory protocols in protocols.io, where a protocol can be assigned its own identifier (DOI) such that it can be cited independently in the future. For instructions see: http://journals.plos.org/plosone/s/submission-guidelines#loc-laboratory-protocols

We look forward to receiving your revised manuscript.

Kind regards,

Fernando A. Wilson, PhD

Academic Editor

PLOS ONE

Journal Requirements:

"Authors GL, LZ, AD, GW, JS, KL, and QZ are employees of Janssen Research & Development, LLC and hold stock in the company. JR is in research collaboration with Janssen, AstraZeneca, Abbvie, and Pfizer, for which Karolinska Institutet has received grant support. JR has been a speaker for Eli Lilly and received unrestricted grant support from Schering-Plough. Wilson Joe, PhD, of MedErgy, provided editorial support for this manuscript. Editorial support was funded by Janssen Research & Development, LLC."

 i) Please confirm that this does not alter your adherence to all PLOS ONE policies on sharing data and materials, by including the following statement: "This does not alter our adherence to  PLOS ONE policies on sharing data and materials.” (as detailed online in our guide for authors http://journals.plos.org/plosone/s/competing-interests).  If there are restrictions on sharing of data and/or materials, please state these. Please note that we cannot proceed with consideration of your article until this information has been declared.

 ii)  Please include your updated Competing Interests statement in your cover letter; we will change the online submission form on your behalf.

Additional Editor Comments (if provided):

- In your revised manuscript, verify that PLOS ONE formatting requirements are addressed. (https://journals.plos.org/plosone/s/submission-guidelines)

- In the text, reference numbers should be enclosed in square brackets (https://journals.plos.org/plosone/s/submission-guidelines#loc-references)

Reviewers' comments:

Reviewer's Responses to Questions

**Comments to the Author**

1. Is the manuscript technically sound, and do the data support the conclusions?

Reviewer #1: Yes

Reviewer #2: No

2. Has the statistical analysis been performed appropriately and rigorously? 

Reviewer #1: Yes

Reviewer #2: No

3. Have the authors made all data underlying the findings in their manuscript fully available?

Reviewer #1: Yes

Reviewer #2: Yes

4. Is the manuscript presented in an intelligible fashion and written in standard English?

Reviewer #1: Yes

Reviewer #2: Yes

5. Review Comments to the Author

Reviewer #1: This study used 2013 Optum Clinformatics claims data to examine the association of treatment-resistant depression with healthcare resource utilization and costs in the U.S. The authors found that patients with treatment-resistant depression were more likely to receive inpatient, outpatient, and emergency care, and to have higher healthcare costs during the follow-up period after the index date.

[1] Major

None.

[2] Minor

1. P.9, line157. The authors may want to perform a sensitivity analysis (i.e., 1:1 propensity score matching), given that they used the propensity score at a 1:4 ratio, and the literature suggests that increasing the number of untreated subjects matched to each treated subject tends to increase the bias in the estimated treatment effect (e.g., Austin PC, Am J Epidemiol. 2010).

2. P.9, line 161. Please clarify whether hospital length of stay (LOS) is total LOS during the follow-up period or LOS per admission.

3. Please clarify the terminology and/or definitions. For example,

(1) P.9., lines 162-167. “Payer” seems to mean insurer and “costs to payer” seems to mean insurance coverage/reimbursement in this context, given that total health care costs are sum of “payer” costs and patients’ out-of-pocket costs. The authors may want to clarify this throughout the text and in Table 3.

(2) P.9., lines 162-167. The authors defined (a) “payer” costs over service types (e.g., outpatient, inpatient, ED), (b) patient cost as the sum of the patient’s out-of-pocket costs (e.g., deductible, copayment), and (c) total healthcare costs as the sum of “payer” and patient costs. This is confusing because (i) both “payers” and patients are types of payers and (ii) both of them pay costs across various service types (i.e., “payer” costs are the amount covered by insurers, and patient costs are the amount not covered by insurers).

(3) Pages 14 and 16, and Tables 2 and 3. The interpretation of the effects of treatment with resistant depression on outpatient visits, hospital LOS, and costs was somewhat unclear (e.g., “estimated mean differences” or “estimate of mean difference”). To improve the interpretability of the estimates, the authors may want to define the effects as average marginal effect (i.e., difference in adjusted predicted outcomes between patients with and without treatment-resistant depression) and use this terminology consistently in the text and tables.

Reviewer #2: Thank you for your paper. As you will see below, I have concerns about this study's operational definition of treatment resistant depression (TRD), the inclusion of both privately insured persons and Medicare Advantage beneficiaries without distinguishing between them in the analyses, and the dissimilarity between the two matched groups in terms of the comorbidity index.

1) Abstract and title: Please include information about the subjects (e.g., persons with private or Medicare Advantage insurance coverage). Payer and patient expenditures would be expected to vary greatly by insurance type, so this is important information.

2) Page 4 paragraph 1: The second sentence indicates that the estimated economic burden of MDD in the US in 2010 was US$210.5 billion, but the last sentence indicates that the societal economic burden of MDD in 2012 was $188 billion. This inconsistency is not explained and thus raises questions. Is MDD becoming less economically burdensome or is this due to methodological differences? Please rework this first paragraph so the reader is not distracted by the inconsistency in past research. (e.g., provide more information, or the last sentence could become less specific, simply stating that the economic burden of MDD has been estimated to exceed that of cancer and diabetes but providing no specific numbers).

3) Page 4-5 lines 67 through 70: You indicate that reference number 10 (by Al-Harbi) proposes that an adequate duration is treatment for >=4 weeks with >=3 weeks on an adequate dose. Reading this reference, it appears that this suggestion was not Al-Harbi's but was put forth in a paper cited by Al-Harbi in their introduction section (reference #9 in that paper - Thase ME, Rush JA. Treatment-resistant depression. In: Bloom FE, Kupfer DJ, editors. Psychopharmacology. New York, NY: Raven; 1995.). However, Al-Harbi goes on to review the literature and ultimately concludes in the Discussion section that "It seems that depression should only be considered drug resistant after at least 6 weeks of two trials of antidepressant therapy" (page 383). Al-Harbi also discusses the 6 week time period in "Optimization of Antidepressants" section on page 374. Thus, your statement that Al-Harbi proposes a >=4 week time period appears to be inaccurate -- >-4 weeks actually contradicts the conclusions of the review.

4) Page 5 lines 70 through 73: You say that your methodology for adequate dose is based on the Massachusetts General Hospital (MGH) Antidepressant Treatment Response Questionnaire (ATRQ). However, you do not use the Massachusetts General Hospital Antidepressant Treatment Response Questionnaire's definition of adequate duration: According to the article you cite (#11), "The MGH ATRQ defines 6 weeks on an adequate dose of antidepressant medication as an adequate duration of treatment." Please change your definition accordingly and rework your analyses. Doing so would be consistent both with the Al-Harbi article you cite, the MGH ATRQ that you reference, and numerous recent studies that used claims data to examine the costs of treatment resistant depression, including but not limited to one recently published in PLOS ONE (see #5 below). Others include Amos et al 2018 "Direct and Indirect Cost Burden and Change of Employment Status in Treatment Resistant Depression"; Benson et al 2020, "An evaluation of the clinical and economic burden among older adult Medicare-covered beneficiaries with treatment resistant depression"; Pilon et al 2019, "Medicaid spending burden among beneficiaries with treatment-resistant depression"; Pilon 2019, "US integrated delivery networks perspective on economic burden of treatment resistant depression: retrospective matched cohort study." This is not a complete list - I am only providing a few examples.

5) Page 6 line 101: Presumably you only include persons 18-63 (rather than 18-64, which is a more typical age range) because you require two years of continuous eligibility during the follow-up period and you wanted to limit the study to working-age adults. However, it is unclear why you limited the age range to working-age adults given that your data source includes both privately insured persons and Medicare Advantage members (according to lines 95-98 on page 6). Over 85% of Medicare Advantage members were >=65 during the period you describe, and previous research indicates that TRD is a burden in the Medicare population >=65 years of age (see Pilon et al 2019, "Burden of treatment-resistant depression in Medicare: A retrospective claims database analysis," PLOS ONE and Benson et al 2020, "An evaluation of the clinical and economic burden among older adult Medicare-covered beneficiaries with treatment resistant depression," Am J of Geriatric Psychiatry). Further, Medicare beneficiaries under age 65 are likely to be unlike the privately insured persons that are included in your data: They're only eligible because they are receiving Social Security Disability Insurance (SSDI) payments or were diagnosed with end-stage renal disease (ESRD) or amyotrophic lateral sclerosis (ALS). Given all of this, please do one of the following: expand your age range or exclude Medicare Advantage beneficiaries from your analysis.

6) Methods: If you expand your age range and retain Medicare Advantage beneficiaries in your analysis, please include the insurance type (MA or private) in the characteristics that were assessed and compared between the two groups. Please also include this in the propensity score matching -- the costs and patient characteristics would be expected to differ greatly for the two groups.

7) Page 6 line 107: Please provide a citation for the UnitedHealthcare Guidelines. It is unusual to include adjustment disorder diagnoses within the major depression disorder diagnostic group - explain/justify.

8) Page 8 line 135-136 - Please include a citation justifying your inclusion of non-antidepressant medications (it is justifiable, but there should be a citation)

9) Page 8 line 137-139 - Please create a supplemental file that defines the specific recommended minimal dosages for each medication. Doing so is consistent with past claims-based research on the same topic, and it enables other researchers to replicate and/or build on your study

10) Page 8 line 149 - your age ranges in this sentence include persons 55-64 (but you excluded 64 year olds) and >=65 year olds (but these persons were not included in the study). Please update this language as needed depending on how you approach the change requested in #5 above.

11) Page 9 line 159 through 163 - Please clarify what is included in "outpatient visits" -- is this ambulatory hospital outpatient visits, office-based visits, or both? Also, clarify "medical claims" given that outpatient, inpatient, ED and pharmacy are listed separately.

12) Page 9 line 164 through 165 - You say that patient costs were the sum of deductibles, copayments and coinsurance and you mention procedures. Are patient costs for prescription pharmaceuticals included? If so, I recommend that you reword: Patient costs were defined as the sum of deductibles, copayments and coinsurance for all medical and pharmacy services and supplies paid through patients' insurance benefits (or something similar). If not, please explain the decision to exclude out of pocket pharmacy costs.

13) Page 8 line 141 through 144: Please see feedback #4 above regarding the definition of "adequate duration." This operational definition is questionable; a change is needed.

14) Page 8 line 150 - patients often have multiple types/categories of depression diagnoses in claims, even in a short period of time. If a person had >1 depression diagnosis within 30 days of the index date, how were they categorized into a single group?

15) Page 7 line 152 - please provide a citation for the version of the Elixhauser comorbidity score that you used

16) Page 11 - the propensity score-matched data differed on Elixhauser score, suggesting that the propensity score matching was not wholly successful in rendering the two groups similar in terms of the important variables associated with costs. This is a significant issue. Please justify the decision to adjust for this variable rather than tightening the matching logic for the propensity score matching, or rework to tighten the matching logic. If not reworked, please include a discussion of this issue in the limitations section.

17) Page 21 line 294 - You describe the high unemployment rates of persons with TRD, but at the same time your sample primarily consists of persons with employer sponsored insurance (as that is what is most prevalent in the Optum data) and you require continuous enrollment in the health plan (and thus you're requiring continuous employment). Please add this to the limitations section -- your study may not be representative of many persons with TRD. Instead, it represents those able to be continuously employed, which may be those with less severe forms of TRD.

18) Page 21 last paragraph - TRD is defined solely on medications and does not take into account other treatment strategies for depression, including ECT, rTMS, VNS, or psychotherapy. Please add this to the limitations section -- not all TRD may be identified based on a medication-only algorithm.

19) General comment on discussion section - it is unclear what your paper adds to the existing literature given the large number of studies that already explore this topic. Please emphasize what is new/different/notable about your study and explain the importance of the new information provided by your study.

20) General comment on discussion/other sections in terms of references - you do not look to many of the most recent articles on the costs of TRD in your discussion and other sections of the manuscript. See a few listed above, and this is not a complete list. Please update your literature review and update your paper accordingly.

6. PLOS authors have the option to publish the peer review history of their article (what does this mean?). If published, this will include your full peer review and any attached files.

Reviewer #1: No

Reviewer #2: No

---

## [Author Response · Author response to Decision Letter 0]

23 Jul 2020

Comments

Journal Requirements

Comment 1: Please ensure that your manuscript meets PLOS ONE’s style requirements, including those for file naming. The PLOS ONE style templates can be found at 

Response: Formatting updates have been made so that the manuscript meets PLOS ONE style requirements.

Comment 2: PLOS requires an ORCID iD for the corresponding author in Editorial Manager on papers submitted after December 6th, 2016. Please ensure that you have an ORCID iD and that it is validated in Editorial Manager. To do this, go to ‘Update my Information’ (in the upper left-hand corner of the main menu), and click on the Fetch/Validate link next to the ORCID field. This will take you to the ORCID site and allow you to create a new iD or authenticate a pre-existing iD in Editorial Manager. Please see the following video for instructions on linking an ORCID iD to your Editorial Manager account: https://www.youtube.com/watch?v=_xcclfuvtxQ.

Response: The corresponding author’s ORCID iD has been verified in Editorial Manager.

Comment 3: We note that you have indicated that data from this study are available upon request. PLOS only allows data to be available upon request if there are legal or ethical restrictions on sharing data publicly. For information on unacceptable data access restrictions, please see http://journals.plos.org/plosone/s/data-availability#loc-unacceptable-data-access-restrictions. 

*In your revised cover letter, please address the following prompts:

Response: There are legal restrictions on sharing the de-identified data set used in our analysis. The data were made available via a third-party license from Optum; Janssen Pharmaceuticals has a license for analysis of the Optum Clinformatics™ Extended Data Mart (CEDM). As such, we cannot provide the raw data themselves. However, other researchers can access the data by purchase through Optum. Those who are interested may visit https://www.optum.com/solutions/life-sciences.html for more information on accessing Optum CEDM data. We confirm that we had no special privileges to access data from Optum via third-party license, and that other researchers would be able to access the data in the same manner.

Comment 4: Thank you for stating the following in the Competing Interests section:

“Authors GL, LZ, AD, GW, JS, KL, and QZ are employees of Janssen Research & Development, LLC and hold stock in the company. JR is in research collaboration with Janssen, AstraZeneca, Abbvie, and Pfizer, for which Karolinska Institutet has received grant support. JR has been a speaker for Eli Lilly and received unrestricted grant support from Schering-Plough. Wilson Joe, PhD, of MedErgy, provided editorial support for this manuscript. Editorial support was funded by Janssen Research & Development, LLC.”

i) Please confirm that this does not alter your adherence to all PLOS ONE policies on sharing data and materials, by including the following statement: "This does not alter our adherence to PLOS ONE policies on sharing data and materials.” (as detailed online in our guide for authors http://journals.plos.org/plosone/s/competing-interests). If there are restrictions on sharing of data and/or materials, please state these. Please note that we cannot proceed with consideration of your article until this information has been declared.

ii) Please include your updated Competing Interests statement in your cover letter; we will change the online submission form on your behalf.

Please know it is PLOS ONE policy for corresponding authors to declare, on behalf of all authors, all potential competing interests for the purposes of transparency. PLOS defines a competing interest as anything that interferes with, or could reasonably be perceived as interfering with, the full and objective presentation, peer review, editorial decision-making, or publication of research or non-research articles submitted to one of the journals. Competing interests can be financial or non-financial, professional, or personal. Competing interests can arise in relationship to an organization or another person. Please follow this link to our website for more details on competing interests: http://journals.plos.org/plosone/s/competing-interests.

Response: We have amended the competing interests section of the manuscript to state that “These interests do not alter the authors’ adherence to PLOS ONE policies on sharing data and materials.”

Editor Comments

Comment 1: In your revised manuscript, verify that PLOS ONE formatting requirements are addressed. (https://journals.plos.org/plosone/s/submission-guidelines)- In the text, reference numbers should be enclosed in square brackets (https://journals.plos.org/plosone/s/submission-guidelines#loc-references).

Response: Formatting updates have been made so that the manuscript meets PLOS ONE style requirements. The revised manuscript uses square brackets, rather than parentheses, to enclose in-text citations.

Reviewer #1 Comments 

Comment 1: P.9, line157. The authors may want to perform a sensitivity analysis (i.e., 1:1 propensity score matching), given that they used the propensity score at a 1:4 ratio, and the literature suggests that increasing the number of untreated subjects matched to each treated subject tends to increase the bias in the estimated treatment effect (e.g., Austin PC, Am J Epidemiol. 2010).

Response: Thank you very much for pointing out a potential concern with our methodology and for the suggestion to use a 1:1 ratio in propensity score matching (PSM). The results for a 1:1 ratio are consistent with those from a 1:4 ratio for PSM. The results of 1:1 ratio matching are presented in three new tables in the Supporting Information (S6, S7, and S8) as the following:

Comment 2: P.9, line 161. Please clarify whether hospital length of stay (LOS) is total LOS during the follow-up period or LOS per admission.

Response: The LOS is the sum of hospital stay days from all hospitalizations during one-year follow-up period. This is now clarified in the Methods section (p 8-9; new text in red font):

“Number of outpatient visits (which included office based and ambulatory hospital outpatient visits), proportion of patients with emergency department (ED) visits, proportion of patients with hospitalizations, and hospital length of stay (LOS; ie, the sum of hospital stay days from all hospitalizations during the one-year follow-up period) were assessed to measure resource utilization.”

Comment 3: Please clarify the terminology and/or definitions. For example:

(1) P.9., lines 162-167. “Payer” seems to mean insurer and “costs to payer” seems to mean insurance coverage/reimbursement in this context, given that total health care costs are sum of “payer” costs and patients’ out-of-pocket costs. The authors may want to clarify this throughout the text and in Table 3.

Response: Thank you for the suggestion for clarifications. The sentences have been revised as follows (p 9): 

“Medical costs to payers included claims for outpatient visits, ED visits, and hospitalizations; pharmacy costs to payers were the sum of pharmacy claims; and total costs to payers were the sum of medical costs and pharmacy costs to payers. Medical costs to patients were defined as the sum of deductibles, copayments, and coinsurance for all medical services; prescription costs to patients were defined as the sum of deductibles, copayments, and coinsurance for all prescription drugs; and total costs to patients were the sum of medical costs and prescription costs to patients. Total healthcare costs were defined as the sum of costs to payers and patients.” 

This text has also been added as a footnote below Table 3 for additional clarity.

(2) P.9., lines 162-167. The authors defined (a) “payer” costs over service types (e.g., outpatient, inpatient, ED), (b) patient cost as the sum of the patient’s out-of-pocket costs (e.g., deductible, copayment), and (c) total healthcare costs as the sum of “payer” and patient costs. This is confusing because (i) both “payers” and patients are types of payers and (ii) both of them pay costs across various service types (i.e., “payer” costs are the amount covered by insurers, and patient costs are the amount not covered by insurers).

Response: Thank you for the suggestion; please see the above response.

(3) Pages 14 and 16, and Tables 2 and 3. The interpretation of the effects of treatment with resistant depression on outpatient visits, hospital LOS, and costs was somewhat unclear (e.g., “estimated mean differences” or “estimate of mean difference”). To improve the interpretability of the estimates, the authors may want to define the effects as average marginal effect (i.e., difference in adjusted predicted outcomes between patients with and without treatment-resistant depression) and use this terminology consistently in the text and tables.

Response: We agree with the Reviewer and have adjusted the language in the text accordingly (please see p 14-18).

Reviewer #2 Comments 

General comment: Thank you for your paper. As you will see below, I have concerns about this study's operational definition of treatment resistant depression (TRD), the inclusion of both privately insured persons and Medicare Advantage beneficiaries without distinguishing between them in the analyses, and the dissimilarity between the two matched groups in terms of the comorbidity index.

Response: Thank you for pointing out the distinction between private versus Medicare Advantage insurance coverage. The focus of this study was on patients aged 18 to 63 years at index (see the abstract, ‘Sample selection and study design’ section, and Table 1). We have clarified this point in the text and limitations section of the Discussion. 

The text in the ‘Assessments’ section (p 8) now reads:

“Characteristics included: age group (18–24, 25–34, 35–44, 45–54, and 55–63 years); sex; index year…”

The Discussion now includes the following limitation (p 22):

“Furthermore, results obtained using the Optum ClinformaticsTM database may not generalize beyond patients with employer-sponsored commercial insurance and Medicare Advantage insurance.”

Additionally, we checked the insurance type for the patients in this study and confirmed that only 5 out of 9289 non-TRD patients had both commercial and Medicare insurance; all TRD patients had commercial insurance.

Comment 1: Abstract and title: Please include information about the subjects (e.g., persons with private or Medicare Advantage insurance coverage). Payer and patient expenditures would be expected to vary greatly by insurance type, so this is important information.

Response: Thank you for the suggestion. As described in the previous response, we have confirmed that all the patients in the study cohort had commercial insurance coverage and only 5 non-TRD patients had additional Medicare Advantage coverage. We do not expect this will change our results substantially. 

The manuscript title has been revised as follows: 

“A retrospective analysis to estimate the healthcare resource utilization and cost associated with treatment-resistant depression in commercially insured US patients”

Additionally, the abstract objective now reads:

“The economic burden of commercially insured patients in the United States with treatment-resistant depression and patients with non–treatment-resistant major depressive disorder was compared using data from the Optum Clinformatics™ claims database.”

Comment 2: Page 4 paragraph 1: The second sentence indicates that the estimated economic burden of MDD in the US in 2010 was US$210.5 billion, but the last sentence indicates that the societal economic burden of MDD in 2012 was $188 billion. This inconsistency is not explained and thus raises questions. Is MDD becoming less economically burdensome or is this due to methodological differences? Please rework this first paragraph so the reader is not distracted by the inconsistency in past research (e.g., provide more information, or the last sentence could become less specific, simply stating that the economic burden of MDD has been estimated to exceed that of cancer and diabetes but providing no specific numbers).

Response: Thank you for bringing this distraction to our attention. The introductory paragraph has been reworked and now reads as follows (p 4):

“Depression is a widespread, severely disabling disorder associated with impaired daily functioning, diminished quality of life, and increased mortality and healthcare utilization [1-4]. Healthcare costs such as outpatient medical services, pharmaceutical services, and inpatient services as well as indirect costs such as workplace presenteeism and absenteeism all contribute substantially towards the total burden of major depressive disorder [3]. In 2012, the US societal economic burden of major depressive disorder was estimated at $188 billion, which exceeded the US societal burden of cancer ($131 billion) and diabetes ($173 billion) [4].” 

Comment 3: Page 4-5 lines 67 through 70: You indicate that reference number 10 (by Al-Harbi) proposes that an adequate duration is treatment for >=4 weeks with >=3 weeks on an adequate dose. Reading this reference, it appears that this suggestion was not Al-Harbi's but was put forth in a paper cited by Al-Harbi in their introduction section (reference #9 in that paper - Thase ME, Rush JA. Treatment-resistant depression. In: Bloom FE, Kupfer DJ, editors. Psychopharmacology. New York, NY: Raven; 1995.). However, Al-Harbi goes on to review the literature and ultimately concludes in the Discussion section that "It seems that depression should only be considered drug resistant after at least 6 weeks of two trials of antidepressant therapy" (page 383). Al-Harbi also discusses the 6 week time period in "Optimization of Antidepressants" section on page 374. Thus, your statement that Al-Harbi proposes a >=4 week time period appears to be inaccurate -- >-4 weeks actually contradicts the conclusions of the review.

Response: There are indeed varying definitions and suggestions for what is an “adequate duration” for a treatment trial. After further critical reading of the Al-Harbi (2012) article, he seems not to have done a thorough review of what duration would be optimal. In fact, it seems somewhat illogical that he refers to both Thase & Rush (1995) and Thase et al. (Thase ME, Blomgren SL, Barkett MA, et al. Fluoxetine treatment of patients with major depressive disorder who failed to initial treatment with sertraline. J Clin Psychiatry. 1997;58:16-21) to support the claim cited above that at least six weeks should be required – because in fact those two references do not at all reflect the full spectrum of studies which have investigated this. 

Another recent, and much more thorough, review is by Gaynes et al. 2020 (Gaynes BN, Lux L, Gartlehner G, et al. Defining treatment-resistant depression. Depress Anxiety. 2020;37(2):134-145). They reviewed a large number of studies and wrote “Experts do not agree on how to define an adequate dose and adequate duration. Typically, the minimum duration cited is 4 weeks.” Further, they write (in section 3.2.2): “…we then determined whether the investigators had confirmed the duration: that is, clarified that patients previously received what KQ 1 [KQ1: What definitions of TRD appear in these sources? Do definitions converge on the best one?] had indicated was an adequate dose. In KQ 1, approximately one-half of the eligible reviews and guidelines identified a minimum of 4 weeks of treatment; the other half identified it as 6 weeks. We defined an adequate dose here as 4 weeks because one primary tool to confirm the adequacy of dose and duration, the ATHF, required at least 4 weeks to be considered as adequate duration. Of 185 studies, 146 (79%) considered in their selection criteria whether the patient had been treated previously with an adequate dose; 112 (61%) systematically confirmed that the dose was adequate by specifying dosage levels. Of all 185 studies, 150 (81%) considered in their selection criteria whether prior treatments were of adequate duration; 128 (69%) systematically confirmed that the duration was adequate (≥4 weeks of treatment).”

In conclusion, we no longer consider Al-Harbi to be a preferred reference for “adequate duration” but instead Gaynes et al. 2020, and we have revised the text accordingly (p 4-5):

“Although no consensus definition currently exists, the US Agency for Healthcare Research and Quality (AHRQ) and Food and Drug Administration (FDA) proposed a standard definition of treatment-resistant depression: failing to respond to a minimum of 2 antidepressants administered at an adequate dose, for an adequate duration [8,9]. There is currently no consensus regarding the definitions of ‘adequate’; a recent review on treatment-resistant depression found that most studies considered an adequate treatment duration to last for a minimum of 4 or 6 consecutive weeks, with the majority requiring ≥4 weeks [10,11]. In the current study, we defined adequate dose based upon the American Psychiatric Association Practice Guidelines for Treatment of Patients with Major Depressive Disorder [12] and approved recommended minimal dosage, and adequate duration was defined by an algorithm that required ≥29 days of prescription coverage (details outlined in Materials and Methods below). Results from STAR*D suggest that nonresponse to 2 adequate trials of established pharmacotherapy classes is an inflection point that predicts a poor prognosis with respect to low remission and high relapse rates, and is associated with higher rates of future medication intolerance [5,13].”

Comment 4: Page 5 lines 70 through 73: You say that your methodology for adequate dose is based on the Massachusetts General Hospital (MGH) Antidepressant Treatment Response Questionnaire (ATRQ). However, you do not use the Massachusetts General Hospital Antidepressant Treatment Response Questionnaire's definition of adequate duration: According to the article you cite (#11), "The MGH ATRQ defines 6 weeks on an adequate dose of antidepressant medication as an adequate duration of treatment." Please change your definition accordingly and rework your analyses. Doing so would be consistent both with the Al-Harbi article you cite, the MGH ATRQ that you reference, and numerous recent studies that used claims data to examine the costs of treatment resistant depression, including but not limited to one recently published in PLOS ONE (see #5 below). Others include Amos et al 2018 "Direct and Indirect Cost Burden and Change of Employment Status in Treatment Resistant Depression"; Benson et al 2020, "An evaluation of the clinical and economic burden among older adult Medicare-covered beneficiaries with treatment resistant depression"; Pilon et al 2019, "Medicaid spending burden among beneficiaries with treatment-resistant depression"; Pilon 2019, "US integrated delivery networks perspective on economic burden of treatment resistant depression: retrospective matched cohort study." This is not a complete list - I am only providing a few examples.

Response: In regards to the first part of the Reviewer’s comment (ie, adequate duration), we appreciate the Reviewer pointing out the inaccurate citation of Chandler et al. 2010; in fact, we did not follow the Massachusetts General Hospital Antidepressant Treatment Response Questionnaire. We apologize for the confusion and have revised the text accordingly (p 4-5):

“In the current study, we defined adequate dose based upon the American Psychiatric Association Practice Guidelines for Treatment of Patients with Major Depressive Disorder [12] and approved…” 

The duration concern is answered in our response to this Reviewer’s comment #3. Briefly, Al-Harbi was not consistent and did not provide a recommendation, mentioning both 4 and 6 weeks. 

To provide some additional context: we started our TRD research in 2016 and explored variations in the definition of ‘adequate duration’ in a feasibility study based on the data available through the end 2015. We set the upper limit as 180 days, i.e., if a patient was on the same treatment for 180 days or fewer, this patient would not be considered to have TRD. This consideration was based on the STAR*D study in which <1% of patients were counted as a treatment failure with a duration of >180 days. We then examined the impact of the choice of the lower limit of the ‘adequate duration’ definition at 14, 28, and 42 days; the result was TRD rates of 11.79%, 10.91%, and 10.17% (please see the following figure). This suggested minimal differences among the choices of the lower limit.

In regards to the second part of the Reviewer’s comment (ie, patient population), this study was an analysis of commercially insured patients (as described in our response to this Reviewer’s comment #1). 

Comment 5: Page 6 line 101: Presumably you only include persons 18-63 (rather than 18-64, which is a more typical age range) because you require two years of continuous eligibility during the follow-up period and you wanted to limit the study to working-age adults. However, it is unclear why you limited the age range to working-age adults given that your data source includes both privately insured persons and Medicare Advantage members (according to lines 95-98 on page 6). Over 85% of Medicare Advantage members were >=65 during the period you describe, and previous research indicates that TRD is a burden in the Medicare population >=65 years of age (see Pilon et al 2019, "Burden of treatment-resistant depression in Medicare: A retrospective claims database analysis," PLOS ONE and Benson et al 2020, "An evaluation of the clinical and economic burden among older adult Medicare-covered beneficiaries with treatment resistant depression," Am J of Geriatric Psychiatry). Further, Medicare beneficiaries under age 65 are likely to be unlike the privately insured persons that are included in your data: They're only eligible because they are receiving Social Security Disability Insurance (SSDI) payments or were diagnosed with end-stage renal disease (ESRD) or amyotrophic lateral sclerosis (ALS). Given all of this, please do one of the following: expand your age range or exclude Medicare Advantage beneficiaries from your analysis.

Response: The focus of this study was on privately insured patients. The age restriction used in the analysis would eliminate Medicaid patients. Additional details on the associated clarifications made in the revised manuscript are described in our responses to this Reviewer’s general comment and comment #1.

Comment 6: Methods: If you expand your age range and retain Medicare Advantage beneficiaries in your analysis, please include the insurance type (MA or private) in the characteristics that were assessed and compared between the two groups. Please also include this in the propensity score matching -- the costs and patient characteristics would be expected to differ greatly for the two groups.

Response: We have confirmed that all the patients included in this cohort study had commercial insurance coverage, as described in more detail in our responses to this Reviewer’s general comment and comment #1.

Comment 7: Page 6 line 107: Please provide a citation for the UnitedHealthcare Guidelines. It is unusual to include adjustment disorder diagnoses within the major depression disorder diagnostic group - explain/justify.

Response: Please see reference #24, which is cited at the end of the sentence in question. This reference is also embedded below and accessible via this link: https://www.uhccommunityplan.com/assets/healthcareprofessionals/pharmacyprogram/FL-Pharmacy/ICD-9_Code-Drug_Match_FL_FHK.pdf.

Comment 8: Page 8 line 135-136 - Please include a citation justifying your inclusion of non-antidepressant medications (it is justifiable, but there should be a citation).

Response: We have added two references, Al-Harbi 2012 and Gelenberg et al. 2010 (references #12 and 25), on page 7. Al-Harbi 2012 lists all these drugs as augmentation therapies in Table 3, and the drugs were also discussed in Gelenberg et al. 2010.

Comment 9: Page 8 line 137-139 - Please create a supplemental file that defines the specific recommended minimal dosages for each medication. Doing so is consistent with past claims-based research on the same topic, and it enables other researchers to replicate and/or build on your study.

Response: Such a table was included in the original submission as supporting information. Please see S1 Table (‘List of antidepressant medications and minimum adequate dose’), which includes the following footnotes: 

“aStarting doses were based on the recommended starting dose indicated in the American Psychiatric Association (APA) Practice Guidelines for Treatment of Patients with Major Depressive Disorder, 3rd edition, 2010 (https://psychiatryonline.org/pb/assets/raw/sitewide/practice_guidelines/guidelines/mdd.pdf).

bStarting doses for other antidepressant medications not included in the APA Practice Guidelines for Treatment of Patients with Major Depressive Disorder were based on the starting doses indicated in the label (http://www.accessdata.fda.gov/scripts/cder/drugsatfda/index.cfm).

cOther selected medications from the database include an antidepressant-antipsychotic combination treatment indicated for treatment resistant depression, a selected antianxiety agent, and other agents not approved for use in the United States (US Food and Drug Administration. Drugs@FDA: FDA-approved drugs. Available from: https://www.accessdata.fda.gov/scripts/cder/daf/).

dNot applicable; not approved for use in the United States.”

Comment 10: Page 8 line 149 - your age ranges in this sentence include persons 55-64 (but you excluded 64 year olds) and >=65 year olds (but these persons were not included in the study). Please update this language as needed depending on how you approach the change requested in #5 above.

Response: This was a typo and has been corrected. The text in the ‘Assessments’ section (p 8) now reads:

“Characteristics included: age group (18–24, 25–34, 35–44, 45–54, and 55–63 years); sex; index year…”

Comment 11: Page 9 line 159 through 163 - Please clarify what is included in "outpatient visits" -- is this ambulatory hospital outpatient visits, office-based visits, or both? Also, clarify "medical claims" given that outpatient, inpatient, ED and pharmacy are listed separately.

Response: We appreciate this suggestion and the following clarifications have been added (please see p 8-9):

• Outpatient visits included office based and ambulatory hospital outpatient visits based on place of service variable in data

• Medical claims included claims for outpatient visits, ED visits, hospitalization by place of service variable in data

• Pharmacy claims were the prescription claims from pharmacies, excluding in-hospital medication administration

Comment 12: Page 9 line 164 through 165 - You say that patient costs were the sum of deductibles, copayments and coinsurance and you mention procedures. Are patient costs for prescription pharmaceuticals included? If so, I recommend that you reword: Patient costs were defined as the sum of deductibles, copayments and coinsurance for all medical and pharmacy services and supplies paid through patients' insurance benefits (or something similar). If not, please explain the decision to exclude out of pocket pharmacy costs.

Response: Thank you; the text has been modified as you suggested. The text has been revised as follows (p 9): 

“Medical costs to payers included claims for outpatient visits, ED visits, and hospitalizations; pharmacy costs to payers were the sum of pharmacy claims; and total costs to payers were the sum of medical costs and pharmacy costs to payers. Medical costs to patients were defined as the sum of deductibles, copayments, and coinsurance for all medical services; prescription costs to patients were defined as the sum of deductibles, copayments, and coinsurance for all prescription drugs; and total costs to patients were the sum of medical costs and prescription costs to patients. Total healthcare costs were defined as the sum of costs to payers and patients.” 

This text has also been added as a footnote below Table 3 for additional clarity.

Comment 13: Page 8 line 141 through 144: Please see feedback #4 above regarding the definition of "adequate duration." This operational definition is questionable; a change is needed.

Response: Please see our response to this Reviewer’s comments #3 and #4.

Comment 14: Page 8 line 150 - patients often have multiple types/categories of depression diagnoses in claims, even in a short period of time. If a person had >1 depression diagnosis within 30 days of the index date, how were they categorized into a single group?

Response: We ordered the 4 categories by their seriousness, from high to low as: major depressive disorder (ICD-9 296.X), dysthymic disorder (ICD-9 300.X), adjustment disorder (ICD-9 309.X), and depressive disorder NOS (ICD-9 311.X). In case a patient had 2 categories, this patient was assigned the more serious category. Few patients had codes in multiple categories.

Comment 15: Page 7 line 152 - please provide a citation for the version of the Elixhauser comorbidity score that you used.

Response: The reference is: Quan H, Sundararajan V, Halfon P, Fong A, Burnand B, Luthi JC, et al. Coding algorithms for defining comorbidities in ICD-9-CM and ICD-10 administrative data. Med Care. 2005;43: 1130-1139. This reference is now cited as reference #26. 

Comment 16: Page 11 - the propensity score-matched data differed on Elixhauser score, suggesting that the propensity score matching was not wholly successful in rendering the two groups similar in terms of the important variables associated with costs. This is a significant issue. Please justify the decision to adjust for this variable rather than tightening the matching logic for the propensity score matching, or rework to tighten the matching logic. If not reworked, please include a discussion of this issue in the limitations section.

Response: Thank you for the great feedback. The small p-value was due to large sample sizes. The mean differences were 0.1 and 0.07 before and after matching, respectively, and were small. In addition, we checked the quality of matching using ‘standardized difference’, which has become a standard metric replacing the p-value approach. A value of the absolute standardized difference <0.1 is considered as comparable between groups by Austin (Austin, PC, Balance diagnostics for comparing the distribution of baseline covariates between treatment groups in propensity-score matched samples, Statist. Med. 2009; 28:3083–3107 (www.interscience.wiley.com) DOI: 10.1002/sim.3697). The following table suggests that TRD and non-TRD were comparable before matching. The matching did improve the balance among most variables as measured by ‘standardized difference’. Nevertheless, we reported results as planned in the study protocol.

Comment 17: Page 21 line 294 - You describe the high unemployment rates of persons with TRD, but at the same time your sample primarily consists of persons with employer sponsored insurance (as that is what is most prevalent in the Optum data) and you require continuous enrollment in the health plan (and thus you're requiring continuous employment). Please add this to the limitations section -- your study may not be representative of many persons with TRD. Instead, it represents those able to be continuously employed, which may be those with less severe forms of TRD.

Response: We appreciate this comment and have acknowledged this limitation with the following statement (p 21):

“Furthermore, results obtained using the Optum ClinformaticsTM database may not generalize beyond patients with employer-sponsored commercial insurance and Medicare Advantage insurance.”

Comment 18: Page 21 last paragraph - TRD is defined solely on medications and does not take into account other treatment strategies for depression, including ECT, rTMS, VNS, or psychotherapy. Please add this to the limitations section -- not all TRD may be identified based on a medication-only algorithm.

Response: We used a medication-based algorithm to mimic the TRD definition that has been applied by the US FDA and EU EMA in their current regulatory reviews of TRD treatments. For patients with TRD, psychotherapy might not be sufficient to control the depressive symptom without an antidepressant treatment, so we did not count psychotherapy alone as a line of treatment. We consider this is a conservative approach. The non-pharmacologic procedures like ECT, rTMS, and VNS, are usually used in the later lines after multiple pharmacologic treatment failures. We expect most of these patients have been captured in our study cohort who have failed at least two antidepressant treatments in the current depression episode.

Comment 19: General comment on discussion section - it is unclear what your paper adds to the existing literature given the large number of studies that already explore this topic. Please emphasize what is new/different/notable about your study and explain the importance of the new information provided by your study.

Response: This is a very good suggestion. We have emphasized an important aspect of our analysis, which is the data on patients’ out-of-pocket healthcare costs. The paragraph in the Discussion section on page 21 now reads:

“In contrast to most previous studies, which focused on healthcare costs reimbursed by payers, the current study also examined patients’ out-of-pocket costs. Patients with treatment-resistant depression had out-of-pocket medical and pharmacy costs of US$1,323 in year 1 and US$1,254 in year 2. Compared with non–treatment-resistant major depressive disorder, costs for treatment-resistant depression represented increases of US$354 in year 1 and US$184 in year 2. These costs are likely to represent a substantial burden for many patients with treatment-resistant depression. In the STAR*D study, participants reported high unemployment rates, ranging from 36% for patients who responded to step 1 treatment to 47% for patients who progressed to step 4 [5]. In another STAR*D analysis, patients with treatment-resistant depression demonstrated lower vocational productivity than patients with non–treatment-resistant major depressive disorder [31]. A claims-based study found that employees with treatment-resistant depression had an average of 35.8 work loss days per year, which was 1.7 times the rate of work loss days in employees with non–treatment-resistant major depressive disorder and 6.2 times that of those without major depressive disorder [14]. Thus, the higher out-of-pocket healthcare costs associated with treatment-resistant depression shown in the current analysis may represent a considerable financial hardship for this vulnerable population.”

Comment 20: General comment on discussion/other sections in terms of references - you do not look to many of the most recent articles on the costs of TRD in your discussion and other sections of the manuscript. See a few listed above, and this is not a complete list. Please update your literature review and update your paper accordingly.

Response: We appreciate this comment and have added recent articles describing costs related to TRD to the Introduction and Discussion sections of our manuscript. These articles are:

20. Pilon D, Sheehan JJ, Szukis H, Singer D, Jacques P, Lejeune D, et al. Medicaid spending burden among beneficiaries with treatment-resistant depression. J Comp Eff Res. 2019;8: 381-392.

21. Pilon D, Joshi K, Sheehan JJ, Zichlin ML, Zuckerman P, Lefebvre P, et al. Burden of treatment-resistant depression in Medicare: a retrospective claims database analysis. PLoS One. 2019;14: e0223255.

22. Pilon D, Szukis H, Joshi K, Singer D, Sheehan JJ, Wu JW, et al. US integrated delivery networks perspective on economic burden of patients with treatment-resistant depression: a retrospective matched-cohort study. Pharmacoecon Open. 2020;4: 119-131.

The Introduction has been updated as follows (p 5):

“According to three recent estimates, per-patient per-year direct healthcare costs in patients with treatment-resistant depression versus non–treatment-resistant major depressive disorder were US$6,709 higher among commercially-insured patients, $4,382 higher among Medicaid-insured patients, and $9,479 higher among US integrated delivery network–insured patients [14,20,22].”

The Discussion has been updated as follows (p 19-20):

“In this study, mean total healthcare costs to payers in year 1 and year 2, respectively, were 45% and 27% higher for patients with treatment-resistant depression compared with those with non–treatment-resistant major depressive disorder. This result is consistent with prior work across a range of patient populations, which found a 25%-134% higher burden among those with treatment-resistant depression versus those with non–treatment-resistant major depressive disorder [14-22].”

---

## [Decision Letter · Decision Letter 1]

26 Aug 2020

A retrospective analysis to estimate the healthcare resource utilization and cost associated with treatment-resistant depression in commercially insured US patients

PONE-D-19-34560R1

Dear Dr. Li,

We’re pleased to inform you that your manuscript has been judged scientifically suitable for publication and will be formally accepted for publication once it meets all outstanding technical requirements.

Kind regards,

Fernando A. Wilson, PhD

Academic Editor

PLOS ONE

Additional Editor Comments (optional):

Reviewers' comments:

Reviewer's Responses to Questions

**Comments to the Author**

1. If the authors have adequately addressed your comments raised in a previous round of review and you feel that this manuscript is now acceptable for publication, you may indicate that here to bypass the “Comments to the Author” section, enter your conflict of interest statement in the “Confidential to Editor” section, and submit your "Accept" recommendation.

Reviewer #1: All comments have been addressed

2. Is the manuscript technically sound, and do the data support the conclusions?

Reviewer #1: Yes

3. Has the statistical analysis been performed appropriately and rigorously? 

Reviewer #1: Yes

4. Have the authors made all data underlying the findings in their manuscript fully available?

Reviewer #1: Yes

5. Is the manuscript presented in an intelligible fashion and written in standard English?

Reviewer #1: Yes

6. Review Comments to the Author

Reviewer #1: [1] Major

None.

[2] Minor

None. All comments have been adequately addressed in the revised text and response letter.

7. PLOS authors have the option to publish the peer review history of their article (what does this mean?). If published, this will include your full peer review and any attached files.

Reviewer #1: No

---

## [Editor Report · Acceptance letter]

1 Sep 2020

PONE-D-19-34560R1 

A retrospective analysis to estimate the healthcare resource utilization and cost associated with treatment-resistant depression in commercially insured US patients 

Dear Dr. Li:

I'm pleased to inform you that your manuscript has been deemed suitable for publication in PLOS ONE. Congratulations! Your manuscript is now with our production department. 

Kind regards, 

on behalf of

Dr. Fernando A. Wilson 

Academic Editor

PLOS ONE